# A Novel Therapeutic Tumor Vaccine Targeting MUC1 in Combination with PD-L1 Elicits Specific Anti-Tumor Immunity in Mice

**DOI:** 10.3390/vaccines10071092

**Published:** 2022-07-08

**Authors:** Jiayi Pan, Wuyi Zeng, Jiangtao Jia, Yi Shi, Danni Wang, Jun Dong, Zixuan Fang, Jiashan He, Xinyu Yang, Rong Zhang, Menghua He, Maoping Huang, Bishi Fu, Bei Zhong, Hui Liu

**Affiliations:** 1School of Basic Medical Sciences, The Sixth Affiliated Hospital of Guangzhou Medical University, Qingyuan People’s Hospital, Guangzhou Medical University, Guangzhou 510182, China; panjiayi06@163.com (J.P.); zeng_wuyi@163.com (W.Z.); jjt13111216713@163.com (J.J.); field99116@163.com (Y.S.); dannyw1414@163.com (D.W.); dongjun2190@163.com (J.D.); fangzixuan213@163.com (Z.F.); hejiashan0527@163.com (J.H.); yangxinyu022@163.com (X.Y.); 13415576603@163.com (R.Z.); 15622149185@163.com (M.H.); ping660815@126.com (M.H.); fubishi@gzhmu.edu.cn (B.F.); 2Clinical Laboratory, Guangdong Provincial People’s Hospital, Guangdong Academy of Medical Science, Guangzhou 510080, China; 3The State Key Laboratory of Respiratory Disease, Guangdong Provincial Key Laboratory of Allergy & Clinical Immunology, Guangzhou 510260, China

**Keywords:** dendritic cells, PD-L1, tumor vaccine, MUC1, immunotherapy

## Abstract

Dendritic cells (DCs), as professional antigen-presenting cells (APCs), play a key role in the initiation and regulation of humoral and cellular immunity. DC vaccines loaded with different tumor-associated antigens (TAAs) have been widely used to study their therapeutic effects on cancer. A number of clinical trials have shown that DCs are safe as an antitumor vaccine and can activate certain anti-tumor immune responses; however, the overall clinical efficacy of DC vaccine is not satisfactory, so its efficacy needs to be enhanced. MUC1 is a TAA with great potential, and the immune checkpoint PD-L1 also has great potential for tumor treatment. Both of them are highly expressed on the surface of various tumors. In this study, we generated a novel therapeutic MUC1-Vax tumor vaccine based on the method of PD-L1-Vax vaccine we recently developed; this novel PD-L1-containing MUC1-Vax vaccine demonstrated an elevated persistent anti-PD-L1 antibody production and elicited a much stronger protective cytotoxic T lymphocyte (CTL) response in immunized mice. Furthermore, the MUC1-Vax vaccine exhibited a significant therapeutic anti-tumor effect, which significantly inhibited tumor growth by expressing a high MUC1^+^ and PD-L1^+^ level of LLC and Panc02 tumor cells, and prolonged the survival of cancer-bearing animals. Taken together, our study provides a new immunotherapy strategy for improving the cross-presentation ability of therapeutic vaccine, which may be applicable to pancreatic cancer, lung cancer and for targeting other types of solid tumors that highly express MUC1 and PD-L1.

## 1. Introduction

It is well known that the immune system plays an important role in controlling tumor growth and development, and the cell-killing ability of tumor-specific T cells is crucial for eradicating tumors in vivo. The activation of T cells requires antigen-presenting cells (APCs), and dendritic cells (DCs) are the most effective APC [1]. Tumor antigens expressed by tumor cells can be taken up by DCs, processed, and presented to T cells and other immune cells, thus activating cellular and humoral immunity and inspiring a strong specific anti-tumor immune response [2,3]. In order to activate tumor-specific T cells more efficiently, tumor antigens of interest have been loaded onto cultured DCs and then reinfused into the body. These “trained” DCs can induce a strong and specific anti-tumor immune response [4]. At present, the anti-tumor effect of DC vaccines has been tested in more than 200 clinical trials, and some of them have achieved good efficacy [5,6]; however, the overall clinical response is still somewhat unsatisfactory, which may be attributed to the insufficient ability of DCs to cross-present antigens.

There are three main ways for DCs to take up antigens: phagocytosis, pinocytosis and receptor-mediated endocytosis [7]; among them, there are many receptors that mediate the cross-presentation of exogenous antigens, with the most common being the Fcγ receptors (FcγRs) that expresses on the surface of DCs. FcγRs can bind to the Fc end of the immunogen-IgG complex (immune complex, ICs), from which the antigen can be effectively captured, processed, and cross-presented to MHC-I and MHC-II by DCs, thus effectively inducing cytotoxic T lymphocyte cells (CTL) and helper T cells (Th); moreover, FcγRs mediated endocytosis enables DCs to activate the immune response more efficiently [8,9,10]. Therefore, this feature can be used to improve the design of DC vaccine antigen uptake, so as to improve the antigen presentation ability of the DC vaccine, improve the immune efficacy of the vaccine, and activate Th and CTL more effectively.

In addition to the mode of DCs uptake affecting the anti-tumor efficacy of the vaccine, the source, and type of tumor antigen it is loaded with also play a key role in tumor immunotherapy. Mucin 1 (MUC1) is a highly glycosylated transmembrane protein containing two structural domains, MUC1-N and MUC1-C. MUC1 is normally expressed on the surface of most glandular epithelia [11,12] but overexpression of MUC1 can be found on many tumors, such as lung, pancreatic, breast, ovarian, and prostate cancers [11,13,14]. The National Cancer Institute (NCI) ranks MUC1 second on its list of 75 tumor-associated antigens, based on criteria such as therapeutic function, immunogenicity, specificity, and carcinogenicity [15]. And so far, many scholars have studied the anti-tumor efficacy of DC vaccine targeting MUC1. Some phase I/II clinical trials have proved that injecting DCs loaded with MUC1 peptide, mRNA or cDNA can effectively inhibit tumor growth, produce a long-lasting anti-tumor response and prolong survival [16,17]; this clearly indicates the potential and clinical application value of MUC1 as a tumor-associated antigen. Unfortunately, these vaccines are not yet in phase III clinical trials.

Programmed death 1 ligand 1 (PD-L1) is a transmembrane protein that is highly expressed on the surface of many tumors, and is also a co-suppressor of immune responses. Highly expressed PD-L1 on tumor cells binds to the PD-1 receptor on the surface of T cells, which reduces the value-added of T cells, inhibits their factor secretion and induces apoptosis, thus leading to immune escape of tumor cells [18,19,20]. It has been shown that the body can activate PD-L1-specific effector T cells to directly target and kill or release cytokines to indirectly kill tumor cells with high PD-L1 expression [21,22]. And PD-L1-specific T cells can release factors after killing tumor cells or lysed tumor cells can release relevant tumor antigens to directly and indirectly enhance other T cell responses, to enhance the efficacy period of immune response, regulate the immunogenicity of DC vaccine, so as to effectively enhance the intensity of immune response [23,24]; moreover, previous evidence showed that DCs or antigen-loaded DCs can directly induce antibody responses and promote proliferation and antibody production of CD40-activated naive and memory B cells during stimulation of B cell responses [25,26]. More recently, our recent studies also found that DCs can activate both CTL and antibody responses [20,27]; therefore, it should be a promising strategy to combine MUC1 with PD-L1 peptide fragment on DCs to activate T cells specifically targeting MUC1 and PD-L1 to target killing MUC1^+^ or PD-L1^+^ tumors.

In this study, we examined the humoral and cellular immune responses of mice immunized with DC vaccine pulsed with a fusion protein consisting of the human MUC1 extracellular region, the PD-L1 extracellular region, and a linkage of IgG Fc fragments (MUC1-Vax). We found that the immunized mice produced a higher persistent anti-PD-L1 antibodies, and activated Th cells and MUC1 and PD-L1 specific CTL. By monitoring the in vivo tumor growth changes, it was demonstrated that this MUC1-Vax vaccine could effectively inhibit the growth of MUC1^+^ or PD-L1^+^ tumor cells; this novel PD-L1-containing MUC1-Vax vaccine provides a novel strategy for generating consistently anti-PD-L1 antibody and T-cell responses for cancer treatment and prevention.

## 2. Materials and Methods

### 2.1. Mice and Cell Lines

Male C57BL/6 mice aged 6 to 8 weeks were purchased from Charles River Laboratories and raised in a specific pathogen-free (SPF) barrier environment at the Animal Center of Guangzhou Medical University. All animal experiments were approved by the Animal Ethics Committee of Guangzhou Medical University. The murine Lewis lung cancer cell line LLC, and the murine pancreatic adenocarcinoma cell line Panc02, were purchased from ATCC (American Model Culture Cell Bank). These cells were transduced with recombinant lentiviral vectors co-expressing MUC1, PD-L1, and puromycin resistance genes to establish tumor cell lines LLC-MUC1-PD-L1 and Panc02-MUC1-PD-L1 stably expressing MUC1 and PD-L1. These tumor cells stably expressing human MUC1 and PD-L1 were treated with 5 μg/mL or 2.5 μg/mL puromycin (Solarbio, Beijing, China) in culture medium.

### 2.2. Protein Production and Purification

To prepare the MUC1-PD-L1 immunogen (MUC1-Vax), a fusion gene of human MUC1 extracellular domain, PD-L1 extracellular domain, T helper epitope sequence and human IgG1 Fc sequence was synthesized and cloned into pET21a expression vector to construct pET-21a-MUC1-PDL1-IgG1 Fc expression plasmid. To express the recombinant protein, it was transformed into an Escherichia coli that could express the endotoxin-free protein (ClearColi^®^ BL21(DE3)) to induce protein expression, and the bacteria were collected and lysed, then the protein was purified with a Ni-NTA agarose column (Abbkine PurKine™ His-Tag Protein Purification Kit, Wuhan, China). After further dialysis, the production and purification steps of these proteins were analyzed by SDS-PAGE and Western blot. The recombinant human PD-L1 protein was purchased from the company Abcam.

### 2.3. Lentiviral-Transduced Tumor Cell Lines

Lentiviral vectors expressing MUC1 and PD-L1 and carrying puromycin resistance genes were constructed using a three-plasmid system (psPAX2, PMD2.G, pHBLV^TM^) for lentiviral packaging. 293T cells were transfected according to the instructions of LipoFiter^TM^ reagent. The supernatant of the virus was collected twice at 48 h and 72 h after transfection, and the lentivirus stock solution was concentrated by centrifugation, and then the titer of the lentivirus was measured. LLC was added with lentivirus venom to infect tumor cells according to MOI = 30 and Panc02 according to MOI = 20. At the same time, the final concentration of 5 μg/mL polybrene was added to improve the infection rate. Fresh medium was replaced after 24 h. After 48 h, LLC and Panc02 were added to a fresh complete medium containing 5 and 2.5 μg/mL puromycin, respectively, for screening of stable cell lines LLC-MUC1-PD-L1 and Panc02-MUC1-PD-L1. Cells were collected after 7 days for further Real-time Quantitative PCR (qPCR), Western blot and flow cytometry analysis.

### 2.4. Preparation of Bone Marrow-Derived Dendritic Cells (BMDCs)

Bone marrow cells from mouse leg bones were washed out with PBS and lysed for erythrocytes, then cultured in a 1640 medium containing 20 ng/mL GM-CSF and 10 ng/mL IL-4 (PeproTech, Cranbury, NJ, USA). The medium was changed every other day and the cytokines were supplemented. On the 7th day of culture, recombinant MUC1-Vax protein, PDL1 protein (100 µg/mL) or PBS was added, and 1 μg/mL LPS was added after 4 h for co-culture for 24 h. BMDCs were detected by flow cytometry, and mice were immunized with antigen-loaded BMDCs with >60% double positivity for CD11c and CD80.

### 2.5. Tumor Model and Vaccination

C57BL/6J mice were randomly divided into 3 groups, and 2 × 10^5^ LLC-MUC1-PD-L1 cells or 3 × 10^6^ Panc02-MUC1-PD-L1 cells in the exponential growth phase were subcutaneously injected into the right flank of the mice. After 7 days of subcutaneous injection of tumor cells, 2 × 10^6^ antigen-loaded dendritic cells were injected through the footpad and immunized twice at weekly intervals. Tumor size was measured every 3 days using vernier calipers. Tumor volume was calculated as (largest diameter) × (shortest diameter) 2 × 0.5 and imaged every 6 days with a small animal live imager (IVIS Lumina XRMS Series III, Perkin Elmer, Boston, MA, USA). The method of the living image was as follows: mice were anesthetized generally, 150 mg D-Luciferin was intraperitoneally injected into mice per kg, and then imaging analysis was performed after 10–15 min of injection.

### 2.6. Flow Cytometry Analysis

The spleen of immunized C57BL/6 mice was made into single-cell suspension, and 2 × 10^6^ cells per sample were used for flow cytometry detection. To detect T cell immune responses in mice, cells were stained with fluorescently labeled antibodies on the surface and intracellularly. Finally, data were collected on the FACS verse (BD Biosciences, Franklin Lakes, NJ, USA) and analyzed with FlowJo software. The antibodies used in this study were as follows: FITC Anti-Mouse CD3 Antibody (Thermo Fisher, Waltham, MA, USA); PE Anti-Mouse CD4 Antibody (BD Biosciences, Shanghai, China); PerCP-Cy5.5 Anti-Mouse CD8 Antibody (BD Biosciences, Shanghai, China); PerCp-Cy5.5 Anti-Mouse CD69 Antibody (Thermo Fisher, Shanghai, China), PE-Cy7 Anti-Mouse IL-2 Antibody (BD Biosciences, Shanghai, China); BV510 Anti- Mouse IFN-γ Antibody (Biolegend, Beijing, China).

### 2.7. Antibody ELISA

Antibody ELISA detection was determined as we described previously [20]. In brief, recombinant PD-L1 protein (0.5 µg/mL) was coated on ELISA plate overnight at 4 °C, blocked with BSA, and then added with serial dilutions of serum, and incubated at room temperature for 2 h. After extensive washing, HRP-labeled anti-mouse IgG antibody (Bioss, Beijing, China) was added and incubated at room temperature for 1 h. Then, TMB was added for color development, and finally, the optical density (OD) was read at 450 nm on a multi-plate reader (Varioskan Flash, Thermo, Kennebunk, ME, USA).

### 2.8. Hematoxylin-Eosin (HE) Staining

Mouse liver and kidney were isolated from each immune group, fixed with 4% paraformaldehyde, embedded in paraffin, and sliced. Then, dewaxing, staining the nucleus with hematoxylin, staining the cytoplasm with eosin, dehydrating and sealing, and then microscopic examination, image collection and analysis were conducted.

### 2.9. Statistical Analysis

Statistical analysis was performed using GraphPad Prism 8.0 statistical software. Animal survival rates were represented by Kaplan–Meier survival curves, and statistical analysis was performed using the log-rank (Mantel–Cox) test. Tumor size was analyzed by two-way analysis of variance (ANOVA), and the remaining data were analyzed by unpaired two-tailed *t*-test. The data shown in the figures are the mean ± standard deviation and represent at least three independent data. A *p* value of <0.05 was considered statistically significant.

## 3. Results

### 3.1. Production of Recombinant MUC1-PDL1-IgG1 Fc Immunogen (MUC1-Vax)

As mentioned in previous studies, MUC1 is a glycoprotein highly expressed on the surface of many tumor cells, and it is a TAA with great potential for tumor-targeted therapy development [28]. PD-L1 is a well-known immune checkpoint and is also highly expressed on various tumor cells. Most of the drugs developed for PD-L1 are a monoclonal antibody. We generated the specific PD-L1-containing MUC1-Vax according to the new method of peptide assembly we recently developed [20]. The fusion gene linking the human MUC1 extracellular region, PD-L1 extracellular region and IgG1 Fc sequence with T helper epitope sequence was synthesized and cloned into Novagen pET21a expression vector to construct the expression vector pET-MUC1-Vax. The Fc peptide of IgG1 was fused to fusion gene to promote DCs to more efficiently uptake and present antigens. The expression vector was then transformed into ClearColi^®^ BL21(DE3) and induced by isopropyl β-D-1-thiogalactoside, so that *E. coli* could express our desired fusion protein MUC1-Vax. Then the cells were collected, lysed, and washed with 2M urea to dissolve the cell proteins, retaining the MUC1-Vax in the form of inclusion bodies, and then using 8M urea to dissolve the inclusion bodies, and then using Ni-NTA agarose column for protein purification. Then the purified MUC1-Vax was dialyzed to replace the protein solvent with PBS. MUC1-Vax fusion protein purified and dialyzed protein samples were analyzed by SDS-PAGE with an expected molecular weight of 65.33 kDa (Figure 1A arrow) and detected by Western blot with anti-His or anti-MUC1 antibodies (Figure 1B,C, arrows); it showed that this MUC1-Vax fusion protein was successfully prepared, which may be used for further studies.

### 3.2. MUC1-Vax-Loaded DC Vaccine Induces Specific T Cell Activation and Cytokine Secretion

The BMDCs required for the vaccine were induced from the bone marrow cells of the mouse leg bone. DCs were cultured with 100 μg/mL of MUC1-PDL1-IgG1 Fc (MUC1-Vax), PD-L1 protein and PBS pulse DCs on day 7, followed by stimulation of cell maturation with LPS. To investigate whether the DC vaccine loaded with MUC1-Vax induced an effective immune response in vivo, tumor-bearing mice were immunized twice with the above DCs at one-week intervals. Four days after the second immunization, the spleens of the immunized mice were isolated to make single-cell suspensions, and the activation of splenic T cells and cytokine secretion in mice were detected by flow cytometry (Figure 2A). CD69 is an early indicator of T cell activation [29], as shown in Figure 2B, the CD69 of CD4^+^ T cells in the MUC1-Vax-DCs group was upregulated compared with the PDL1-DCs group and the PBS-DCs group, implying a higher degree of T cell activation in mice (Figure 2B). After activation, T cells release cytokines such as IFN-γ and IL2 to promote the proliferation of Th1 and CTL, stimulate macrophages and amplify the immune effect [30,31,32]. IFN-γ and IL2 released by CD4^+^T cells and IFN-γ released by CD8^+^T cells in the MUC1-Vax-DCs group were higher than those in the PDL1-DCs and PBS-DCs groups (Figure 2C–E). These data suggest that the MUC1-Vax-DC vaccine-elicited enhanced IFN-γ and IL-2 secretion and induces specific CTL responses; this means that this novel MUC1-Vax-DC vaccine can more broadly activate T lymphocytes and enhance the secretion of IFN-γ and IL-2, resulting in better potential to trigger CTL responses.

### 3.3. Anti-PD-L1 Antibody Response Induced by MUC1-Vax-Loaded DC Vaccine

According to previous studies, DCs can retain or concentrate antigens on the cell surface for a long time for B cells to recognize and then stimulate humoral immune responses and induce immune memory [33,34]. Therefore, we extracted blood from the eyes of tumor-bearing mice immunized twice with DC vaccine and measured the level of PD-L1 antibody in the serum of the mice by ELISA. As shown in Figure 3A, the level of anti-PD-L1 antibody (IgG) in the serum of mice in the MUC1-Vax-DCs immunized group was significantly higher than that in the other two immunized groups (PBS-DCs group and PDL1-DCs group), and the serum level of anti-PD-L1 antibody in mice immunized with PDL1-DCs was also higher than that in the PBS-DCs group; this shows that the DC vaccine loaded with tumor antigen can activate humoral immune response more effectively, and the vaccine loaded with MUC1-Vax can produce a higher level of PD-L1 antibodies compared with that loaded with PDL1 antigen alone, which shows that the MUC1-Vax-DC vaccine is more effective and can stimulate stronger humoral immunity compared with the traditional DC vaccine. To explore whether the vaccine can cause adverse pathological immune responses, we performed histological sections and H&E staining of liver and kidney in mice treated with PBS-DCs, PDL1-DCs and MUC1-Vax-DCs. No obvious inflammatory reactions were found after analysis (Figure 3B), indicating that this novel MUC1-Vax vaccine has no pathological toxicity to immunized mice and is a relatively safe and effective tumor vaccine.

### 3.4. Inhibitory Effect of MUC1-Vax-DC Vaccine on Tumor Growth

Next, we further tested the inhibitory effect of this novel MUC1-Vax vaccine on MUC1^+^ and PDL1^+^ tumor growth in mice. LLC-MUC1-PDL1 and Panc02-MUC1-PDL1 tumor cell lines were inoculated into the right flank of the 6–8 weeks C57BL/6 mice, LLC-MUC1-PDL1 and Panc02-MUC1-PDL1 tumor cell lines for qPCR assay was shown in Appendix A. After 7 days, the mice that successfully grew tumors were randomly divided into 3 different groups, and then the DCs pulsed with MUC1-Vax, PDL1 or PBS were reinfused via footpad injection, and immunized twice, once a week. The tumor size of the mice was then measured every few days with a vernier caliper. As shown in Figure 4A, MUC1-Vax DC vaccine significantly inhibited the growth of LLC-MUC1-PDL1 tumors, PDL1-DCs had a weak inhibitory effect on tumor growth, while PBS-DCs had only a weak or even no inhibitory effect on tumor growth. Notably, in the pancreatic cancer mouse model inoculated with Panc02-MUC1-PDL1, the tumors of the tumor-bearing mice immunized with PBS-DCs, PDL1-DCs and MUC1-Vax DCs appeared to vary degrees of tumor reduction. Especially in Panc02-bearing-mice immunized twice with MUC1-Vax DCs, showed significant tumor shrinkage, while tumors in PBS-DCs and PDL1-DCs groups shrank only slightly or not at all (Figure 4B). And comparing the PDL1-DCs and PBS-DCs groups, the MUC1-Vax DCs group of both tumor-bearing mice significantly improved the survival rate and prolonged the survival period of the tumor-bearing mice (Figure 4C,D). In addition to measuring with vernier calipers, we used a small animal imager to measure the tumor luciferase every 6 days to observe the tumor growth in mice. As shown in Figure 5A,B, the tumor size of the mice in the MUC1-Vax-DCs treatment group was significantly smaller than that in the PDL1-DCs treatment group at the same time point, even one LLC-bearing-mouse had complete tumor regression, and the tumor size in the PDL1-DCs treatment group was also slightly smaller than that of the PBS-DCs treatment group. Repeated experiments yielded similar results. Taken together, these results indicate that this novel PD-L1-containing MUC1-Vax-DC vaccine can inhibit the growth of MUC1^+^ and PD-L1^+^ tumor cells, and may be an effective therapeutic vaccine against MUC1^+^ and PD-L1^+^ tumors.

## 4. Discussion

As a professional APC, DCs play a key role in the initiation and regulation of humoral and cellular immunity. DCs can activate CD4^+^ and CD8^+^ T cells by cross-presenting antigens, indirectly kill tumor cells through Th-secreted cytokines and directly kill tumor cells through the cytolysis of CD8^+^; moreover, DCs can directly or indirectly promote the growth and differentiation of B cells by secreting cytokines, so as to promote humoral immune response [25,35]. The antibodies produced can inhibit or kill tumor cells in a variety of ways. Thus far, the DC vaccine has been widely used to study its therapeutic effect on cancer. More than 200 clinical trials have shown that DCs as an anti-tumor vaccine are safe and can activate certain anti-tumor immune responses, but the clinical response is unsatisfactory, which may be attributed to the insufficient ability of DCs to cross-present antigens [5,36,37]. Therefore, in this study, the targets of our vaccine were selected MUC1 and PD-L1, two molecules highly expressed on many kinds of tumor cells, in order to activate a sufficient amount of MUC1 or PD-L1 specific T cells to kill tumor cells more widely and comprehensively. At the same time, The Fc segment of IgG1 was added to our MUC1-Vax fusion protein to improve the efficiency of DCs in antigen uptake, processing and cross-presentation, and the efficacy of the vaccine was tested in our constructed tumor-bearing mouse models of lung and pancreatic cancer that stably expressed MUC1 and PD-L1. Our study provides a new immunotherapy strategy for improving the cross-presentation ability of DC vaccines and targeting numerous tumors with high expression of MUC1 and PD-L1, such as invasive lung cancer, pancreas, prostate, epithelial ovarian cancer, platinum-resistant tumors, primary lung cancer and breast cancer.

MUC1 is a very promising tumor target, and the immune checkpoint PD-L1 also has great potential for tumor treatment. There are many therapeutic drugs targeting MUC1 and PD-L1, such as MUC1-based cancer vaccines including mRNA vaccine [16], DNA vaccine, viral vaccine, DC vaccine [17] and glycopeptide vaccine, and there are many anti-PD-L1 monoclonal antibodies, such as Atezolizumab, Avelumab and Duravulumab [38,39]. A small number of tumor patients treated with tumor vaccine against MUC1 or anti-PD-L1 monoclonal antibody alone have good clinical efficacy, but there are still many patients who do not respond to such single-target targeting or checkpoint blockade therapies, or whose efficacy is short-lived and unable to stimulate a durable anti-tumor immune response and strong tumor control and regression. At present, there is currently no specific tumor vaccine that combines the two to target and kill tumor cells; therefore, our study is the first time to combine MUC1, PD-L1 and IgG1 Fc to prepare MUC1-Vax fusion protein and load it on DCs to prepare MUC1-Vax vaccine, which is different from the traditional tumor vaccine against MUC1 or anti-PD-L1 alone.

DCs stimulating an antibody response is believed to be a consequence of CD4^+^ Th function, and previous evidence supports that DCs can directly induce antibody responses [25,26]. DCs produce factors, such as IL-10 and IL-6, and express membrane/soluble proteins to promote B cell growth and differentiation [34,40], DCs also produce IL-12 to boost Th1 development and promote B cells to develop humoral responses [41]. Our recent studies demonstrated that DCs can activate both CTL and antibody responses [20,27]. In this study, we found that DCs, loaded with this new MUC1-Vax immunogen can induce the body to continuously produce anti-PD-L1 antibodies, which can inhibit the binding of PD-L1 on tumor cells and PD-1 on T cells, thus relieving the inhibition of tumor cells on T cells to a certain extent and releasing part of T cells. In addition, PD-L1 antibodies can also kill tumor cells through complement-dependent cytotoxicity (CDC), antibody-dependent cytotoxicity (ADCC) and T cell function regulation. In addition to antibody production, the vaccine can also activate Th and MUC1 and PD-L1 specific CTL, and release IL2 and IFN-γ, thereby inhibiting the growth of MUC1^+^ and PD-L1^+^ tumors. In our study, although compared with the PBS-DCs vaccine group, tumor growth of PDL1-DCs immunized mice was inhibited to a certain extent, and T cell activation and cytokines secretion also increased. Interestingly, the MUC1-Vax-DCs vaccine group was significantly more effective than the PDL1-DCs vaccine group in inducing anti-tumor humoral and cellular immune responses and inhibiting tumor cell growth, and had a stronger inhibitory effect on pancreatic and lung cancer cell growth, and the survival time of mice in this group was also longer. In addition, no obvious inflammation and intoxication were found in the liver and spleen H&E staining analysis of the immunized mice. We believe that this novel PD-L1-containing MUC1-Vax vaccine is an effective and relatively safe therapeutic vaccine against MUC1^+^ and PD-L1^+^ tumors, which can get rid of the disadvantages of single checkpoint inhibitor treatments as invalid, since it provokes the body’s stronger anti-tumor immune response, thus controlling and regressing the tumor.

Although the MUC1-Vax vaccine has shown considerable prospects in anti-tumor therapy, we are aware of some limitations of this study. First of all, the design of the experimental control group is not perfect enough and the number of experimental mice is not enough. In the future, it is still necessary to further design a perfect control group and design a single animal survival experiment to verify the efficacy of the vaccine and eliminate the influence of individual differences as much as possible. Furthermore, in vivo antibody levels should be measured at more than three different time points, not one time point, to verify whether the vaccine-induced humoral immune response is durable and effective; moreover, we also need to optimize the sequence of MUC1 epitopes to improve the immunogenicity and explore more DC vaccine modification methods or combined with other treatments to improve the clinical efficacy of the vaccine. Among them, the most commonly used combination therapy is combined immune checkpoint inhibitors. For example, the most common monoclonal antibody against PD-1, anti-PD-1 monoclonal antibody (mAb) can bind PD-1 on T cells. Combined with DC vaccine, anti-PD-1 mAb may produce synergistic, activating T cells in a more specific and effective manner [42,43]. In addition to the combination of immune checkpoint inhibitors, DC vaccines can also be combined with chemotherapy, the traditional therapy, and this combination has achieved unexpected enhancement of anti-tumor immunity in several clinical trials [44]. Studies have shown that the use of cyclophosphamide, temozolomide (TMZ), or gemcitabine as chemotherapy agents before DCs treatment can delay tumor growth and prolong survival in tumor-bearing mice compared with monotherapy [42]. Besides, many scholars have achieved certain efficacy by combining DCs with radiotherapy and targeted drug therapy [45,46]. Therefore, we intend to further study the combined treatment of this novel PD-L1-containing MUC1-Vax tumor vaccine with immune checkpoint inhibitors, such as PD-1 mAb to verify whether it can achieve better efficacy. Follow-up verification should follow the principle of single-variable to set up the control group more strictly, and the combination treatment should be considered whether the mice will have pathological inflammatory reactions. Not only the H&E section of liver and kidney should be done, but also the detection of inflammatory markers, so as to detect the side effects of the combination treatment more sensitively. In addition, our MUC1-Vax-DC vaccine has some potential clinical applications and challenges; therefore, it is of great significance to test whether this novel human MUC1-Vax-DC vaccine can induce anti-PD-L1 antibody and CTL reaction in cancer patients, and to test the safety of this vaccine.

In summary, our data provide, for the first time, a new therapeutic vaccine against MUC1+ and PD-L1+ tumors, and provoke an enhanced anti-tumor immune response, thus controlling and regressing the tumor; this novel PD-L1-containing MUC1 tumor vaccine may have theoretical and clinical value in the application in tumor immunotherapy in human.

## Figures and Tables

**Figure 1 vaccines-10-01092-f001:**
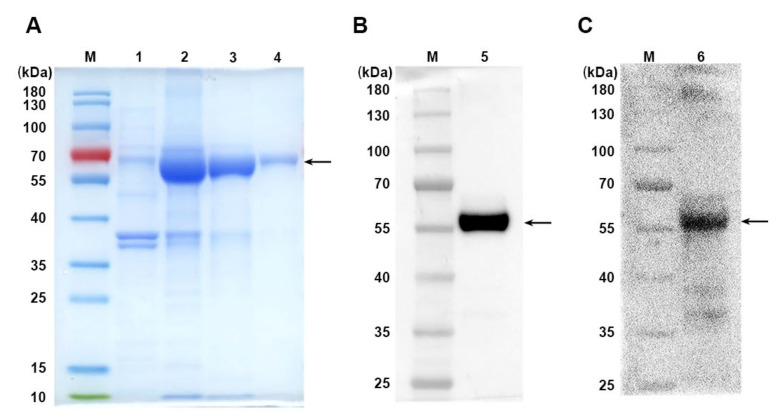
Expression and purification of recombinant protein MUC1-PDL1-IgG1 Fc (MUC1-Vax). (**A**) Plasmid-transfected *E*. coli lysed protein fractions were run on a 12% SDS-PAGE gel and stained with Coomassie Brilliant Blue R250. Lane M is the prestained protein molecular weight marker, lane 1 is the bacterial protein before induction, lane 2 is the bacterial protein after isopropyl β-D-1-thiogalactoside (IPTG) induction, and lane 3 is the purified MUC1-Vax protein, and lane 4 is the MUC1-Vax protein after dialysis. Recombinant protein MUC1-Vax (arrow) is indicated. (**B**) Western blot analysis of purified recombinant protein MUC1-Vax with anti-human His primary antibody. (**C**) Western blot analysis of purified recombinant protein MUC1-Vax with anti-human MUC1 antibody.

**Figure 2 vaccines-10-01092-f002:**
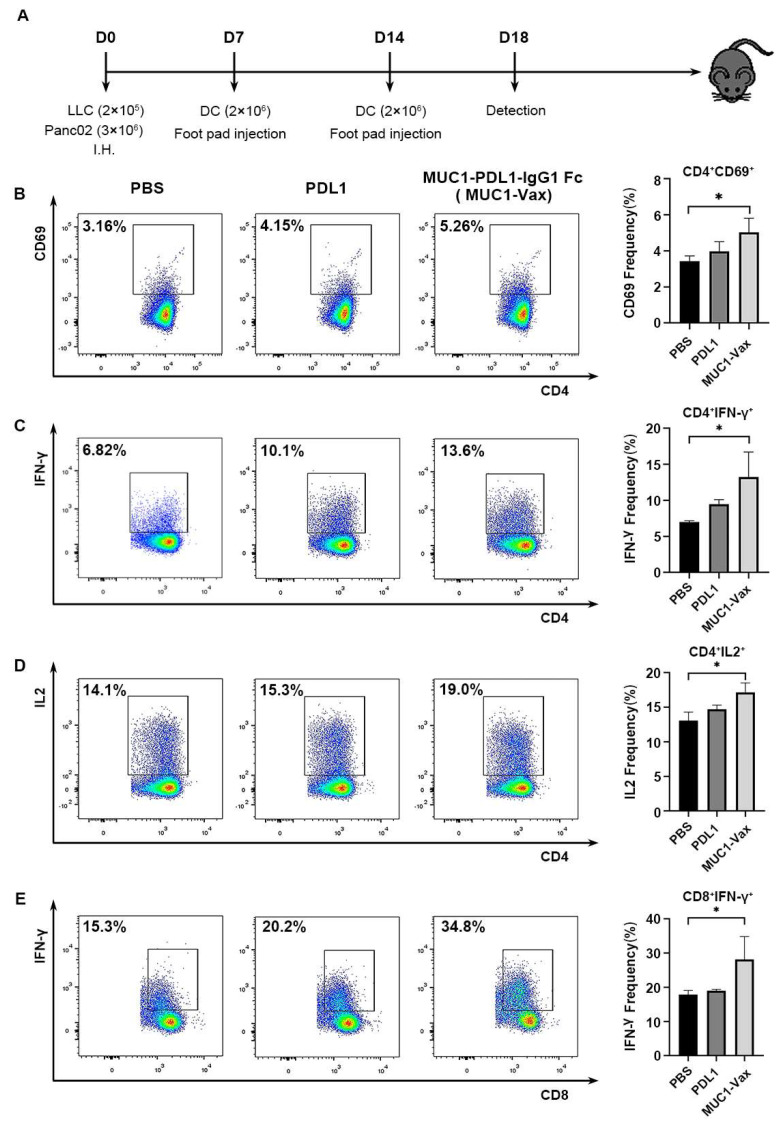
(**A**) C57BL/6 mice were subcutaneously inoculated with 2 × 10^5^ luciferase expressing LLC (LLC-MUC1-PDL1-Luc) or 3 × 10^6^ Panc02 (Panc02-MUC1-PDL1-Luc) cells by hypodermic injection (I.H.). The footpads were injected with 2 × 10^6^ BMDCs loaded with different proteins on the 7th and 14th days after tumor cell injection, respectively. Splenocytes of immunized mice were isolated 4 days after immunization and the activation of CD4^+^ T cells and CD8^+^ T cells (3 per group) and cytokine secretion were detected by flow cytometry. The tumor size, growth, and survival time of the remaining mice were observed (4–5 in each group). (**B**) Activation of CD4^+^ T cells, data are expressed as mean ± SD. (**C**) Secretion of IL-2 by CD4^+^ T cells, data are expressed as mean ± SD. (**D**) Secretion of IFN-γ by CD4^+^ T cells, data are presented as mean ± SD. (**E**) Secretion of IFN-γ by CD8^+^ T cells, data are presented as mean ± SD. * *p* < 0.05.

**Figure 3 vaccines-10-01092-f003:**
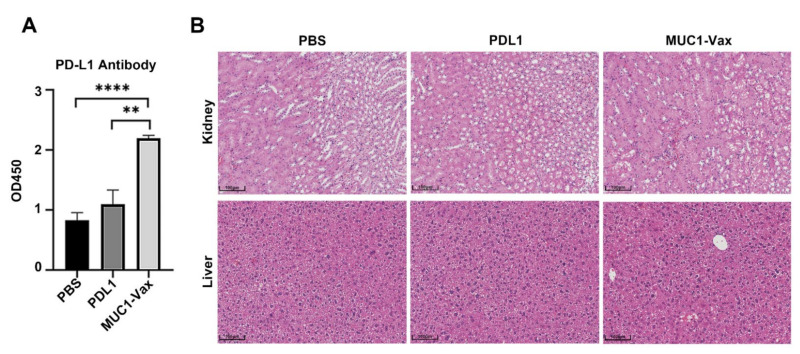
Whole blood, liver and kidney of mice were collected after 4 days of DC vaccine immunization, and serum was separated. (**A**) The serum levels of anti-PD-L1 IgG antibody in different groups were detected by ELISA, ** *p* < 0.01, **** *p* < 0.0001. (**B**) H&E staining analysis of liver and kidney sections. The original magnification of these pictures was ×20.

**Figure 4 vaccines-10-01092-f004:**
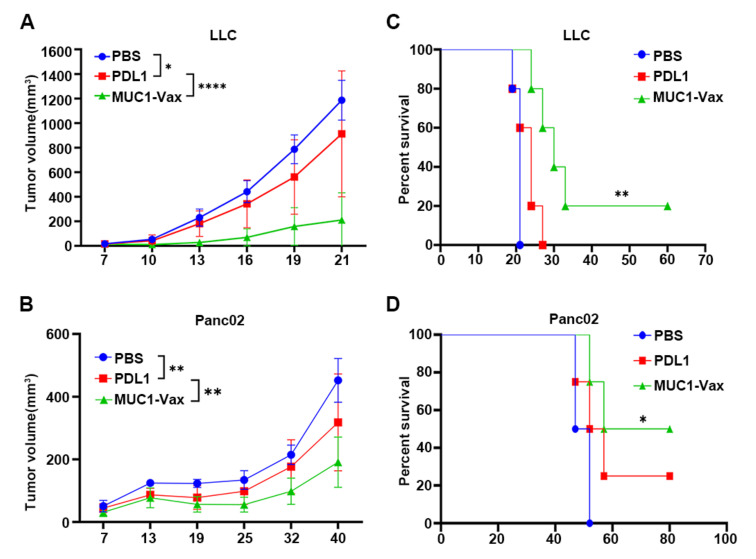
The tumor size of the tumor-bearing mice was measured with a vernier caliper every few days. tumor volume was monitored by the caliper is shown (**A**,**C**). The survival curves are shown (**B**,**D**). There are 4–5 mice in each group, * *p* < 0.05, ** *p* < 0.01, **** *p* < 0.0001.

**Figure 5 vaccines-10-01092-f005:**
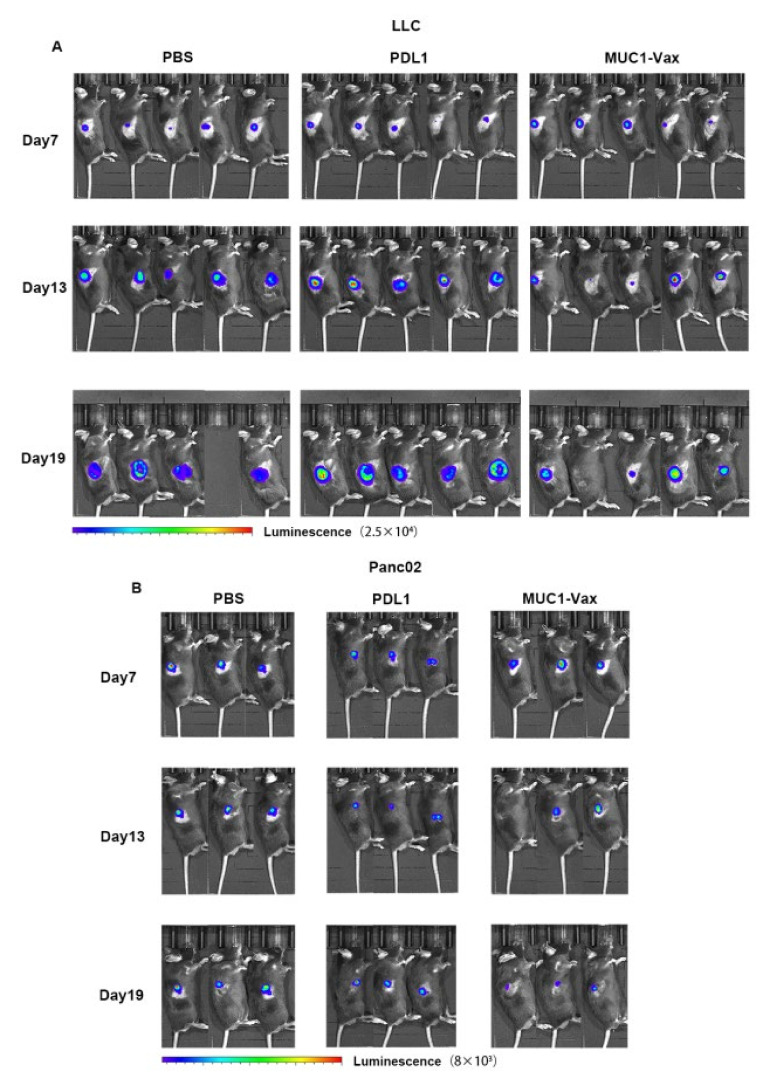
Cell bioluminescence monitoring was performed in vivo on tumor mice inoculated with LLC-MUC1-PDL1-Luc every 6 days, as shown in figure (**A**) Bioluminescence in vivo of mice inoculated with LLC (*n* = 5), blank space means the mouse died; (**B**) In vivo bioluminescence of mice inoculated Panc02 (*n* = 3).

## Data Availability

The data are all available in the paper and online.

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
