# Peer review of "A Novel Therapeutic Tumor Vaccine Targeting MUC1 in Combination with PD-L1 Elicits Specific Anti-Tumor Immunity in Mice"

_vaccines, 2022, doi:10.3390/vaccines10071092_

Round 1
Reviewer 1 Report
Manuscript ID: vaccines-1755384
Title: A Novel Therapeutic Tumor Vaccine Targeting MUC1 in Combination with PD-L1 Elicits Specific Antitumor Immunity in Mice
Authors: Jiayi Pan , Wuyi Zeng , Jiangtao Jia , Yi Shi , Danni Wang , Jun Dong , Zixuan Fang , Jiashan He , Xinyu Yang , Rong Zhang , Menghua He , Maoping Huang , Bishi Fu , Bei Zhong , Hui Liu.
In the present work, the authors generated a novel therapeutic MUC1-Vax tumor vaccine based on a method (PDL1-Vax vaccine) previously developed by themselves. Te immunotherapy strategy combine MUC1, PD-L1 and IgG1 Fc to prepare MUC1-Vax fusion protein and load it on DCs to prepare MUC1-Vax vaccine. Tumor-bearing mice, implanted with LLC and Panc02 tumor cells, were treated with the PD-L1-containing MUC1-Vax vaccine. Animals had an elevated persistent anti-PD-L1 antibody production and elicited a stronger protective cytotoxic T lymphocyte (CTL) response, inhibited the tumor growth, and prolonged the survival. The authors concluded that this new immunotherapy strategy improves the cross-presentation capacity of DC in tumors with high expression of MUC1 and PD-L1
Some questions and comments for the authors:
Line 8 says: Programmed cell death 1 ligand 1 (PD-L1) is…; should read: Programmed death‑ligand 1 (PD‑L1).
Line 100 says MUC1-Vax DC vaccine, which were mediated by … Please define in advance what the Vax DC vaccine is.
Line 115 says: These cells were transduced with recombinant… Transduced or transfected? Please use the correct term.
Lines 115 and 116 say: recombinant lentiviral vectors co-expressing MUC1, PD-L1 and Luciferase… Please explain the origin of these vectors.
Lines 118-119 say: … tumor cells stably expressing human MUC1 and PD-L1treated with 5ug/mL or 2.5ug/mL puromycin (Solarbio, China) in culture medium. Although it is known what the inhibitory effect of puromycin is on protein synthesis, please explain what it was used for in your design of transfected tumor cells.
Lines 150-151 say: …mice were immunized with antigen-loaded BMDCs with >60% double positivity for CD11c and CD80. How did the authors confirm the load efficiency of the BMDCs? Only CD11c and CD80 were used to define the phenotype of DCs? MHC-Class II and FCΥR should have been included, at least.
Lines 153-155: What is the validated antecedent to implant the number of tumor cells referred?
Figure 2A:
1. Were specific T cell activation and cytokine secretion analyzed only in mice implanted with the tumor cell line LLC?
2. What does I.H. mean?
Line 260 says: Therefore, we extracted blood from the eyes… Was the eye punctured to obtain blood samples? Perhaps the correct thing is that the orbital venous plexus was punctured.
Lines 355-360: several citations are written in superscript. It must be corrected.
Lines 375-376 say: …in the liver and the spleen immunohistochemical sections of the immunized mice. This is not correct. H&E staining analysis of liver and kidney sections was carried out (Figure 3), but not immunohistochemistry.
Lines 390-392 have a statement that merits several references: “Studies have shown that the use of cyclophosphamide, temozolomide (TMZ), or gemcitabine as chemotherapy agents before DCs treatment can delay tumor growth and prolong survival in tumor-bearing mice compared with monotherapy.” Please add the appropriate citations.
Other questions:
1. Why were the primary tumors not histopathologically analyzed?
2. The splenic lymphocytes analyzed would reflect the functional status of the lymphocytes that infiltrate tumors (TALs)?
3. The tumor-bearing mice had no metastasis? This aspect had to be analyzed to know if the novel therapeutic tumor vaccine prevents or decreases metastases.
Reviewer 2 Report
The authors explore the potentiality of PD-L1-containing MUC1-Vax tumor vaccine in inhibiting the growth of LLC and Panc02 tumor cells expressing high MUC1+ and PD-L1+ levels.
This study provides an interesting strategy to generate an anti-PD-L1 antibody and T-cell responses for cancer treatment and prevention. The results are interesting. Overall, the work is well executed. I support the publication of this manuscript. However, I have some minor comments that I recommend addressing before publication.
The authors should provide a high magnification image of Fig. 3.
The MOC1-Vax vaccine is effective on murine lung and pancreatic tumor cell lines expressing high MUC1+ and PD-L1+ levels. This is a critical point for the proposed strategy. The expression levels of the two targets should be better defined in order to predict vaccine efficacy. The authors should provide clinical date on lung and pancreatic tumors expressing both high levels of PD-L1 and MUC1.
Reviewer 3 Report
The authors developed a PD-L1-MUC1-Vax vaccine for a possible novel antitumor therapy. It is known that dendritic cells loaded with tumor-associated antigens can be used as antitumor vaccines. An important function could be shown for PD-L1 expression on dendritic cells, which could be used for a better antitumor immune response. Both PD-L1 and MUC1 are overexpressed in some tumors. How good is a combined vaccine in terms of anti-cancer therapy?
The measurement method for determining the tumor volume by measuring the tumor size with a caliber is imprecise and causes a large statistical error. This is known and can be looked up in publications, e.g. 10.1186/1471-2342-8-16. The results should therefore be interpreted with caution. In Figure 4, the tumor volume is given in mm3, but the measurement is made only by determining the largest and the shortest diameter as described in 2.5 Tumor model and vaccination. The evaluation of the fluorescence reporter with an in-vivo imager is a method that should also be used to determine the tumor size.
The study to determine animal survival cannot prove the survival benefit described because the number of animals studied was too small. From the results obtained, an experiment on animal survival can be planned that will show the expected survival advantage with sufficiently high statistical power.
In conclusion, the results cannot prove a "more effective" antitumor therapy.
Reviewer 4 Report
A very well written manuscript on an important research area.
Major comments:
- A schematic illustration of the constructs should be provided. Otherwise, the description of the vaccine leave open some questions regarding its specific structure.
- It would be important to include in the material and method section the protocol used for assessing the absence of bacterial toxins in the vaccine preparations. This is a critical issue considering the immunological potential of bacterial toxins.
- The absence of control vaccine preparations is a concern.
- The LLC and pancreatic cell lines were transfected with expression vectors to obtain cells stably expressing cDNA for MUC-1 and PD-L1. A concern is whether the expression levels obtained in these lines are physiological, i.e. that it compares with levels normally observed in primary tumor cells.
- Although the figure 1 shows a good protein purification of the constructs, there is no structural data that validates its expected structure.
- Figure 2: The data showing the effect of the vaccine on T cell activation on splenocytes is not convincing. In fact, there is no signifcaitn difference between mice injected with PD-L1 alone or with the vaccine, raising serious doubt about the efficacy of the MUC1-Vax.
- Figure 2: The figure legend states that mice were injected with either LLC or Pan02 cells. Yet, there is no indication whether the data shown in figures 2b-e were generated with mice injected with LLC or Pan02.
- Figure 3a: The levels of specific IgG was higher in mice immunized with the vaccine compared to mice injected with PD-L1 alone. This was done at a single time point. It would have been important to determine whether mice injected with the vaccine can maintained higher sustained levels of IgG to rule out the possibility that the vaccine induces a higher but only transient antibody response compared to the PD-L1 control.
- Figure 3b,c: The data shown to rule out the presence of inflammatory reactions in kidneys and liver is solely based on a low-magnification of histological data. This approach is not sufficient to rule out local inflammatory reactions. There is a wealth of inflammatory markers that are more sensitive and that could have been used to address this issue.
- I was not convinced by the data on primary tumor growth induced by the vaccine. This was particularly troubling in the case of the Pan02 model using only 3 mice. This significantly dampened my enthusiasm regarding the potential of this vaccine.
Minor comments:
Line 136: lentivirus stock liquor?
Figure 2: The threshold levels on the x axis on the FCM histograms raise some questions and concerns as they may actually impact on the results and data. There are no gating histograms (FS versus SSC) to validate the gating strategy used to carry out these FCM analyses.
Round 2
Reviewer 4 Report
Although the manuscript was improved, I didn't in the rebuttal how they specifically addressed these issues and which modifications they did to the original ms. Many issues (weaknesses) could have been addressed in the discussion. This would have been very useful for the readers.
This is the reason why I didn't accept the revised version.
Author Response
Thanks for your suggestion, and your point is well taken. We have revised the manuscript in the “discussion section” in the revised manuscript, which would be useful for the readers.
Please see the attachment
